# Non-Psychoactive Cannabinoid Modulation of Nociception and Inflammation Associated with a Rat Model of Pulpitis

**DOI:** 10.3390/biom13050846

**Published:** 2023-05-16

**Authors:** Elana Y. Laks, Hongbo Li, Sara Jane Ward

**Affiliations:** 1Department of Prosthodontics, School of Dentistry, Indiana University, Indianapolis, IN 46202, USA; elaks@iu.edu; 2Center for Substance Abuse Research, Department of Neural Sciences, Lewis Katz School of Medicine, Temple University, Philadelphia, PA 19140, USA; hongboli@temple.edu

**Keywords:** cannabinoids, pulpitis, dental pain

## Abstract

Despite advancements in dental pain management, one of the most common reasons for emergency dental care is orofacial pain. Our study aimed to determine the effects of non-psychoactive *Cannabis* constituents in the treatment of dental pain and related inflammation. We tested the therapeutic potential of two non-psychoactive *Cannabis* constituents, cannabidiol (CBD) and β-caryophyllene (β-CP), in a rodent model of orofacial pain associated with pulp exposure. Sham or left mandibular molar pulp exposures were performed on Sprague Dawley rats treated with either vehicle, the phytocannabinoid CBD (5 mg/kg i.p.) or the sesquiterpene β-CP (30 mg/kg i.p.) administered 1 h pre-exposure and on days 1, 3, 7, and 10 post-exposure. Orofacial mechanical allodynia was evaluated at baseline and post-pulp exposure. Trigeminal ganglia were harvested for histological evaluation at day 15. Pulp exposure was associated with significant orofacial sensitivity and neuroinflammation in the ipsilateral orofacial region and trigeminal ganglion. β-CP but not CBD produced a significant reduction in orofacial sensitivity. β-CP also significantly reduced the expression of the inflammatory markers AIF and CCL2, while CBD only decreased AIF expression. These data represent the first preclinical evidence that non-psychoactive cannabinoid-based pharmacotherapy may provide a therapeutic benefit for the treatment of orofacial pain associated with pulp exposure.

## 1. Introduction

Dental pain affects the quality of life of many individuals. For example, pulpitis, or inflammation of the inner tooth due to decay, physical damage to the tooth, or certain dental procedures, is experienced by 1 in 4 adults between the ages of 20 and 64. Standards of care can range from sealing the tooth to a root canal, tooth extraction, and/or medication. Untreated pulpitis can be associated with severe pain. It has been reported that approximately 75% of those experiencing significant pain associated with pulpitis used non-narcotic analgesics such as ibuprofen or acetaminophen, while nearly 25% used opioid analgesics such as codeine or hydrocodone [1]. Although the authors report significantly better analgesic effects from opioid medications, the dental community has increasingly recognized the need to reduce the prescription of opioids to their patients. Two decades ago, dentists accounted for approximately 15 percent of all immediate-release opioid prescriptions; by 2012, they only wrote 6.4 percent of such prescriptions. To continue this trend of further improving the treatment of dental pain and making it safer, additional effective analgesics are needed.

One such avenue of exploration is focused on understanding the therapeutic potential of cannabinoid-based pharmacotherapies in the treatment of pain and inflammation. The analgesic properties of *Cannabis* have been recognized since ancient times, and in the recent past, our understanding of the multitude of interactions between chemicals derived from *Cannabis* and our endogenous pain and inflammation pathways has vastly expanded. The mammalian endogenous cannabinoid system plays a substantial role in pain homeostasis [2], and exogenous cannabinoids have been shown to be effective in treating a wide range of experimental pain models, including neuropathic and inflammatory pain [3]. Cannabinoid CB_1_ receptor density is high in brain regions involved in pain perception but also in areas associated with mood, perception, and cognition, which associates CB_1_ receptor activation with adverse effects that limit its clinical utility. CB_2_ receptors, in contrast, are predominantly located on the non-neuronal cells of the nervous system and therefore likely selectively target inflammation and other cellular functions while leaving learning and memory, emotional regulation, and other higher order nervous system functions largely unaltered. Therefore, the potential analgesic effects of the mixed CB_1_/CB_2_ receptor agonist phytocannabinoid Δ^9^-tetrahydrocannabinol (Δ^9^-THC) can dampen pain perception and inflammation but are associated with euphorigenic and other CNS effects that hamper clinical use. However, several *Cannabis* constituents are devoid of activity at the CB_1_ receptor and/or interact with additional receptors and signaling molecules outside of the cannabinoid CB receptors, such as GPR55, PPARs, 5-HT receptors, and cytokine and chemokine receptors and ligands, which have all been implicated in the development and maintenance of inflammatory and/or neuropathic pain.

The endocannabinoid system has been shown to be involved in orofacial pain specifically. In a rodent model of trigeminal nerve injury, for example, levels of the endocannabinoid degrading enzyme monoacylglycerol lipase (MAGL) were significantly increased in the trigeminal nuclei. The inhibition of MAGL by JZL-184 led to a significant reduction in injury-associated orofacial sensitivity and a concomitant increase in endocannabinoid levels, suggesting that increasing cannabinoid tone centrally can ameliorate trigeminal nerve pain [4]. Limited work has also been described investigating effects of phytocannabinoid treatment in rodent models of orofacial pain. Wanasuntronwong et al. [5] investigated the efficacy and mechanisms of the non-CB_1_/CB_2_ acting phytocannabinoid cannabidiol (CBD) on orofacial nociception induced by complete Freund’s adjuvant (CFA) injected into the masseter muscle in mice. They found that CFA-induced spontaneous pain-like behaviors were inhibited by the administration of CBD and that this effect was mediated in part by activity at the vanilloid receptor 1 (TRPV1). To our knowledge, however, the impact of *Cannabis*-derived constituents on pulpitis-associated pain and inflammation is unknown.

The goal of the present study is to test the antinociceptive and anti-inflammatory effects of two distinct *Cannabis* constituents, CBD and β-caryophyllene (β-CP), in a rat model of tooth pulp exposure. CBD is one of the most abundant phytocannabinoids found in *Cannabis*. While it binds with relatively low affinity to CB_1_ and CB_2_ receptors, we and others have shown it produces antinociceptive effects in rodent models of inflammatory and neuropathic pain, likely through non-CB receptor mechanisms [6,7,8,9]. β-CP is a sesquiterpene and major component in the essential oils of *Cannabis* as well as other botanicals, which include black pepper and clove, and is recognized as a generally recognized as safe (GRAS) food additive by the U.S. Federal Drug Administration. β-CP has been reported to be a selective CB_2_ receptor agonist with Ki = 155 ± 4 nM [10] and exhibits anti-inflammatory and analgesic properties without producing the intoxicating side effects characteristic of CB_1_ receptor agonists [11]. To investigate the effects of these two *Cannabis* constituents on pulpitis-associated pain and inflammation, sham or left mandibular molar pulp exposures were performed on Sprague Dawley rats. Rats were treated with either vehicle, CBD (5 mg/kg i.p.), or β-CP (30 mg/kg i.p.) administered 1 h prior to pulp exposure, 24 h post-exposure, and on days 3, 7, and 10 post-exposure. Orofacial mechanical allodynia was evaluated at baseline and post-pulp exposure by probing the orofacial region on the ipsilateral and contralateral drilled sides with von-Frey filaments. The animals were then euthanized 15 days post-surgery and the trigeminal ganglia were harvested for the evaluation of gene expression.

## 2. Materials and Methods

The study was approved by the Temple University Animal Care and Use Committee (ACUP 4905). All animals were treated in such a way, where there was no unnecessary pain and discomfort inflicted upon the animal, as outlined by the United States Public Health Service Policy on Humane Care and Use of Laboratory Animals and the Guide for the Care and Use of Laboratory Animals. Prior to the pulp exposure procedure, forty male Sprague Dawley rats were group-housed under a reverse 12 h light/dark cycle with ad libitum access to food and water. The rats were habituated to their surroundings in the vivarium for a period of 7 days. They were then habituated to daily handling and facial sensitivity testing (see below) for an additional week prior to the tooth pulp exposure procedure. Rats were assigned to one of four treatment groups: sham control + vehicle-treated, drilled + vehicle-treated, drilled + CBD-treated, or drilled + β-CP-treated.

### 2.1. Tooth Pulp Exposure Procedure

Following the acclimation period, the rats were anesthetized under isoflurane anesthesia, and one coronal pulpotomy per animal was performed as previously described [12]. Briefly, the rats were reclined at an approximate 45-degree angle, and flexible nose cones were placed over the noses of the rats to deliver isoflurane anesthesia. This allowed for the opening of the jaw to proceed with the drilling procedure. As rats are obligate nose breathers, the surgical plane of anesthesia is maintained without the cone covering the mouth. The mandibular left first molar pulp was exposed using a high-speed drill with ½ round carbide dental bur and left exposed without any further treatment. The animals were observed during recovery from anesthesia and daily for a period of 15 days. They were single-housed from this time on to monitor feeding activity and were also weighed daily. Hydrogel was provided in addition to water and standard chow during the length of the study.

### 2.2. Orofacial Sensitivity Testing

Rats were tested for their withdrawal threshold to the mechanical stimulation of the orofacial skin. To establish baseline withdrawal thresholds, three behavioral sessions on successive days were completed to provide baseline values before the pulp exposure procedure. Each rat was then tested at days 1, 7, and 14 after surgery. Escape responses to mechanical stimulation of the orofacial skin 2–4 mm to the side of the corner of the mouth ipsilateral and contralateral to the drilled molar were assessed while each rat was held loosely in the lap of the investigator. Von Frey filaments of progressively greater rigidity, starting at 1 g and progressing to 60 g, were used. The filament was gently applied to the orofacial region until the filament bent into a c-shape. The filament was held in place for 6 s or until the rat withdrew its head from the stimulus. The procedure involves progressively testing filaments that are thicker in size to induce an escape response. The next lowest filaments were then tested until the rat did not respond to the filament. The thinnest filament that elicited a response was considered the threshold. The testing duration was 15 min for each side. The animals were then euthanized 15 days post surgery, and the trigeminal ganglia were harvested for evaluation of gene expression.

### 2.3. Drug Treatment

The animals were treated with vehicle, CBD (5 mg/kg), or β-CP (30 mg/kg) via intraperitoneal injections. Treatments were administered 1 h prior to pulp exposure, 24 h post-exposure, and on days 3, 7, and 10 post-exposure. Doses of CBD and β-CP were selected from the literature as well as our extensive work with these compounds in other pain models in mice and rats, which demonstrate that these as effective doses of the two compounds respectively. CBD was purchased from Cayman Chemical Company (Ann Arbor, MI, USA) and β-CP was purchased from Sigma-Aldrich (St. Louis, MO, USA). CBD and βC-P were dissolved in a mixture of ethyl alcohol, Cremophor, and saline in a 1:1:18 (*v*/*v*) ratio.

### 2.4. RT-PCR of Trigeminal Ganglia

The trigeminal ganglia of both the ipsilateral and contralateral drilled left lower molar sides in male Sprague Dawley rats were collected and stored in a −80 °C freezer until ready to use. The Quick-RNA MiniPrep Kit (Genesee Scientific Corp, San Diego, CA, USA) was used to extract mRNA, and mRNA was reverse-transcribed into cDNA with the RT2 First Strand Kit (QIAGEN, Hilden, Germany). The TaqManTM PCR primer/probes (Thermo Fisher, Waltham, MA, USA) were used to detect the gene expression of CCL2, CXCL9, TLR4, NLRP3, AIF-1 and GFAP in trigeminal ganglia, and Rn45s (the gene for 45S pre-rRNA transcript) was also detected as a control gene. The quantitative real-time PCR (qRT-PCR) was performed via the StepOnePlus real-time PCR system, and the 2^−ΔΔCT^ method was used to analyze the relative changes in gene expression. The fold changes of target gene expression in trigeminal ganglia were compared to identify the differences between treatment groups.

### 2.5. Statistical Analyses

The three baseline allodynia tests for each rat were averaged to establish a baseline sensitivity score for each rat. To determine the effect of the pulp exposure procedure on orofacial sensitivity, a one-way ANOVA was used to determine the effect of drilling versus sham on ipsilateral and contralateral orofacial sensitivity. Dunnett’s multiple comparison test was used to determine the day(s) in which orofacial sensitivity was significantly different from baseline. To determine the effect of cannabinoid treatment on orofacial sensitivity, a two-way ANOVA was used to determine the effects of treatment, time, and interaction. Tukey’s multiple comparison test was used to determine, for each day, which treatment groups were significantly different from sham control and from drilled + vehicle. To determine the effect of pulp exposure and cannabinoid treatment on gene expression changes in the ipsilateral or contralateral trigeminal ganglia, a one-way ANOVA was used to determine the effect of treatment on gene expression. Dunnett’s multiple comparison test was used to determine the treatment(s) in which gene expression was significantly different from sham or drilled + vehicle.

## 3. Results

Left molar pulp exposure led to a significant increase in mechanical sensitivity on the ipsilateral side of the exposure (Figure 1A). One-way ANOVA revealed a main effect of exposure [F(3,36) = 7.517]. Dunnett’s multiple comparisons test revealed that mechanical thresholds were significantly lower on days 1, 7, and 14 post-pulp exposure (*p* < 0.05). The contralateral side was not affected by the pulp-exposure, and sham rats also showed no alternations in mechanical sensitivity compared with their baselines (Figure 1B–D). These data are presented as raw mechanical threshold values to demonstrate the level of baseline and post-pulp exposure sensitivity. Subsequently, to analyze the effects of cannabinoid treatments on this behavior, data are transformed into percent baseline due to the variations in baseline responding in the groups, as seen in Figure 1A,B.

Cannabinoid treatment produced a significant effect on mechanical threshold for the ipsilateral side Figure 2A). Two-way ANOVA revealed a significant main effect of treatment, [F(3,90) = 17.05], and Tukey’s multiple comparisons test revealed that the drilled + vehicle-treated group showed significantly more mechanical sensitivity than the sham treated rats on days 1, 7, and 14, as shown in Figure 1 (*p* < 0.05). Post hoc analysis also showed a significant difference between the drilled + vehicle-treated versus the drilled + β-CP-treated groups on days 7 and 14 (*p* < 0.05). The CBD-treated group appeared to show a partial effect of treatment in that their mechanical threshold was not significantly different from the sham or drilled-vehicle control. No treatment produced an effect on mechanical responses on the contralateral side (Figure 2B).

The effects of pulp exposure and cannabinoid treatment on the expression of markers of inflammation were also statistically analyzed using one-way ANOVA. The examination of the expression of AIF, the gene that encodes the macrophage/microglial marker Iba-1, in trigeminal ganglia on the ipsilateral drilled side revealed a significant effect of treatment [F(3,14) = 5.995] (Figure 3A). Dunnett’s multiple comparison test showed that pulp exposure (drilled + veh) produced a significant increase in AIF expression compared with sham treatment (*p* < 0.05). Additionally, treatment with either β-CP or CBD significantly downregulated AIF expression in the trigeminal ganglia of the ipsilateral drilled molar side. The examination of the expression of CCL2 (MCP-1, a neuroimmune chemokine implicated in nociception) in the trigeminal ganglia on the ipsilateral drilled side also revealed a significant effect of treatment [F(3,14) = 12.71] (Figure 3B). Dunnett’s multiple comparison test showed that pulp exposure (drilled + veh) produced a significant increase in CCL2 expression compared with sham treatment (*p* < 0.05). β-CP (30 mg/kg) treatment significantly downregulated CCL2 expression compared with the vehicle-treated drilled group. There was a non-significant trend for CBD to also decrease CCL2 expression (*p* = 0.09). This model of pulpitis did not produce any significant changes in the ipsilateral trigeminal ganglia in the expression of TLR-4, CXCL9, NLRP3, or GFAP (Figure 3C–F).

The examination of the expression of AIF in the contralateral trigeminal ganglia also revealed a significant effect of treatment [F(3,13) = 11.45] (Figure 4A). Dunnett’s multiple comparison test showed that pulp exposure produced a significant decrease in AIF expression in vehicle, CBD-, and β-CP-treated groups compared with sham treatment (*p* < 0.05). The examination of the expression of CCL2 in the trigeminal ganglia on the ipsilateral drilled side also revealed a significant effect of treatment [F(3,13) = 4.359] (Figure 3B). Dunnett’s multiple comparison test showed that pulp exposure (drilled + veh) produced a significant increase in CCL2 expression compared with sham treatment (*p* < 0.05). There was also a significant effect of treatment in the contralateral trigeminal ganglia on the expression of CXCL9, a proinflammatory chemokine produced by macrophages and glial cells that has been associated with inflammation in the dorsal root ganglia (DRG) [F(3,13) = 5.536], shown in Figure 4C. Dunnett’s multiple comparison test showed that pulp exposure (drilled + veh) produced a significant decrease in CXCL9 expression in the contralateral trigeminal ganglia (*p* < 0.05). No significant changes in the contralateral trigeminal ganglia in the expression of TLR-4, NLRP3, or GFAP were observed (Figure 4D–F).

## 4. Discussion

The goal of the present study was to determine the behavioral and neuroimmune effects of two distinct, non-psychoactive *Cannabis* constituents, CBD and β-CP, in a rodent model of pulpitis. This is important due to the persistent presence of the opioid crisis and a concomitant dearth of emerging safe and effective analgesics. Considering the rapid global advancements in cannabinoid science and attitude changes regarding the therapeutic use of *Cannabis* products, dentistry has increasingly recognized the need to explore the evidence base surrounding the therapeutic effects of *Cannabis*-based products in the orofacial region (for review, see [13]). To our knowledge, the present work is the first report on the effects of any *Cannabis*-derived constituents on pulpitis-associated nociception and inflammation. In the present model, the pulpotomies performed on the left mandibular first molars produced significant orofacial mechanical allodynia that was present 24 h post-procedure and lasted the duration of the 14-day experimental protocol. This mechanical allodynia was associated with significant elevations in the expression of AIF, the gene that encodes the macrophage/microglial marker Iba-1, and CCL2 (aka MCP-1), a neuroimmune chemokine implicated in nociception. The model was not associated with elevations in CXCL9, TLR-4, NLRP3, or GFAP.

Our present results showed that β-CP significantly attenuated the development of orofacial mechanical allodynia on the ipsilateral drilled molar side. Moreover, examination of the trigeminal ganglia on the ipsilateral drilled side revealed that β-CP significantly attenuated the downregulated expression of AIF. As mentioned, β-CP has previously been described as a CB2 receptor agonist [10]. β-CP exerts anti-inflammatory effects in several other preclinical models, both CB2-receptor-dependent [14] and PPARγ-dependent [15] (for review, see [16]). CB2 receptor agonism has been demonstrated by many to suppress macrophage/microglial activation, both in vitro [17] and in pain models [18]. Similarly, PPARγ has been linked to reduction in neuroinflammation via microglia in models of neuropathic pain [19]. Importantly, recent research has demonstrated that macrophages and microglia participate in neuroinflammation in the trigeminal ganglia following pulp exposure, suggesting that this may play a critical role in the development of inflammatory pain associated with pulpitis [20]. The present study does not investigate the receptor mechanism involved in the β-CP suppression of AIF gene expression; however, mounting evidence suggests that CB2 receptor/PPARγ cross-talk is the critical target for β-CP’s anti-inflammatory effects [21]. Clinically, two CB2 receptor agonists, AZD1940 and GW842166, have been tested in human trials as a single pretreatment to third molar extraction [22,23]. Neither study resulted in significant findings regarding reduction in post-operative pain. Significant differences between this clinical study and our results include the dental pain model, the compound, and the dosing regimen. Taken together, our data demonstrate that β-CP can attenuate macrophage/microglial activation in the trigeminal ganglia concomitant by reducing the development of orofacial allodynia in this rodent model of pulpitis.

The present results with β-CP also add to the growing literature implicating CCL2 in the development or maintenance of inflammatory pain, including orofacial pain associated with pulpitis. The examination of the trigeminal ganglia on the ipsilateral drilled side revealed that β-CP significantly attenuated the upregulated expression of CCL2. In the literature, the administration of CCL2 has been shown to induce painful mechanical hypersensitivity [24], and CCL2 levels are increased in the dorsal root ganglia in rodent models of chronic inflammatory pain, e.g., [25]. CCL2 has been reported to be increased in the tooth pulp following pulpitis [26], but we are unaware of any reports measuring CCL2 expression in the trigeminal ganglia in a rodent model of pulpitis. CCL2 expression in the trigeminal ganglia was increased in a rodent orofacial neuropathic pain model by chronic constriction injury of the infraorbital nerve [27]. It is unclear from our study whether the systemic administration of β-CP prevented local inflammation at the site of the tooth pulp, thus preventing an inflammatory response in the trigeminal ganglia, or whether the effect of β-CP was mediated at the level of trigeminal neurons and/or macrophages/microglia.

While the administration of CBD did not lead to a significant reduction in orofacial sensitivity compared with drilled + vehicle-treated rats, the CBD-treated group was also not significantly more sensitive than the sham controls. Moreover, treatment with CBD was associated with a significant reduction in AIF expression but not a reduction in CCL2. We and others have published extensively on the anti-allodynic effects of CBD in other rodent models of chronic inflammatory or neuropathic pain [6,7,8,9,28]. Since only one dose of CBD was tested in the present study, we cannot conclude that CBD would be ineffective using a different dosing regimen; however, the dose of 5.0 mg/kg used in the present study is within the dose ranges most commonly reported in the literature. In fact, in several studies, this was higher than the minimal effective dose reported. Interestingly, a clinical study assessing the effects of orofacial topical CBD administration in patients with temporomandibular joint disorder showed a significant improvement in VAS for pain compared with the placebo group, as well as a significant improvement in the EMG of bilateral masseter muscles [29]. In summary, these results taken together demonstrate that CBD as a treatment strategy for orofacial pain needs to be further explored.

In summary, our findings in a preclinical rat model of pulpitis associated with orofacial pain add to the wealth of preclinical evidence supporting the use of *Cannabis*-based products for treating chronic nociceptive and neuropathic pain. To our knowledge, this is the first report on the impact of non-psychoactive *Cannabis*-derived constituents on pulpitis-associated pain and inflammation in a rodent model. The sesquiterpene β-CP significantly attenuated orofacial pain and trigeminal ganglia inflammation, while the phytocannabinoid CBD only attenuated a marker of macrophage activation. Further research is warranted to investigate the efficacy of these products and their applicability to this field.

## Figures and Tables

**Figure 1 biomolecules-13-00846-f001:**
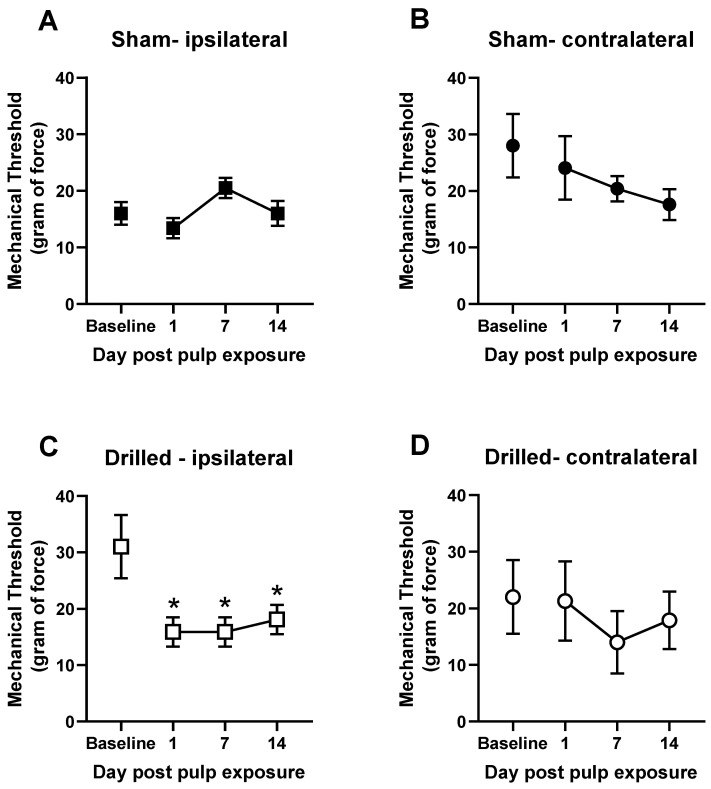
Exposure of the tooth pulp in the left mandibular molar of rats produces ipsilateral orofacial allodynia for up to 2 weeks. Effect of sham procedure or pulp exposure on mechanical withdrawal threshold on the ipsilateral side (**A**,**C**) or contralateral side (**B**,**D**). Orofacial sensitivity was measured with Von Frey filaments prior to pulp exposure procedure and again 1, 7, and 14 days post-exposure. N = 10/group. Ipsilateral and contralateral data are collected from the same rats. * *p* < 0.05 compared with baseline.

**Figure 2 biomolecules-13-00846-f002:**
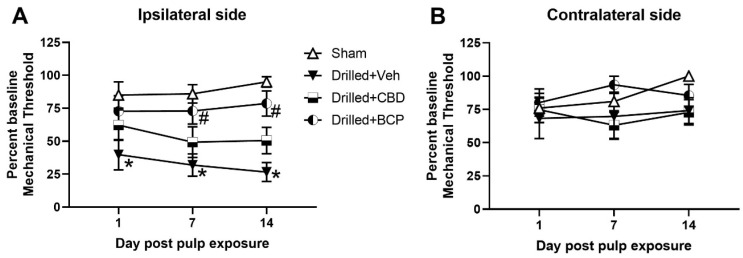
Repeated administration of β-CP significantly attenuated orofacial sensitivity in pulp-exposed rats. Effect of repeated administration (1 h prior to pulp exposure, 24 h post-exposure, and on days 3, 7, and 10 post-exposure) of CBD (5.0 mg/kg) or β-CP (30.0 mg/kg) on mechanical withdrawal threshold on the ipsilateral side (**A**) or contralateral side (**B**). β-CP significantly attenuated orofacial mechanical allodynia on the ipsilateral drilled molar side (* *p* < 0.05 compared with sham, # *p* < 0.05 compared with drilled + vehicle, N = 10/group).

**Figure 3 biomolecules-13-00846-f003:**
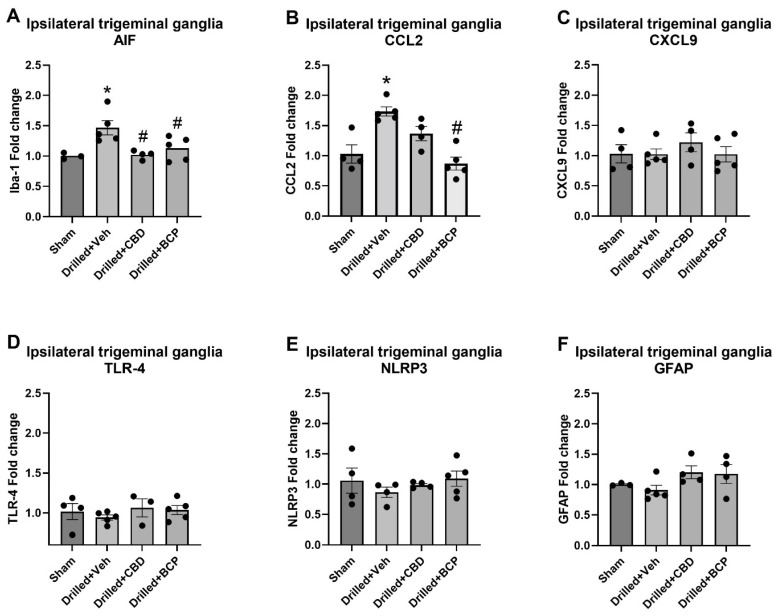
Treatment with *Cannabis* constituents attenuates increases in AIF and CCL2 gene expression in the ipsilateral trigeminal ganglia following pulp exposure in rats. Gene expression was measured using RT-PCR on day 15 post pulp exposure or sham procedure. Only AIF and CCL2 were significantly upregulated. Repeated treatment with β-CP significantly attenuated expression of AIF and CCL2, while CBD only attenuated expression of AIF. * *p* < 0.05 compared with sham, # *p* < 0.05 compared with drilled + vehicle.

**Figure 4 biomolecules-13-00846-f004:**
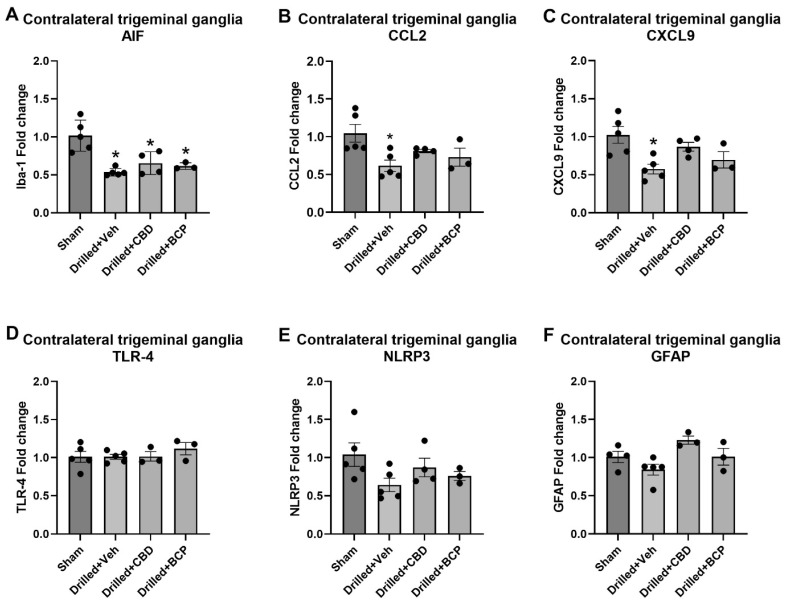
Pulp exposure is associated with a significant decrease in AIF, CCL2, and CXCL9 gene expression in the contralateral trigeminal ganglia in rats. Gene expression was measured using RT-PCR on day 15 post pulp exposure or sham procedure. * *p* < 0.05.

## Data Availability

The data presented in this study are available on request from the corresponding author.

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
