# Peer review of "Non-Psychoactive Cannabinoid Modulation of Nociception and Inflammation Associated with a Rat Model of Pulpitis"

_biomolecules, 2023, doi:10.3390/biom13050846_

Round 1

Reviewer 1 Report

In this study, the authors have tested the behavioral and neuroimmune effects of two non-psychotropic cannabis sativa derived cannabinoids, cannabidiol (CBD) and beta-caryophyllene (β-CP) in a rat model of pulpitis. The authors found β-CP and not CBD to attenuate the development of orofacial mechanical allodynia on the side ipsilateral to the drilled molar teeth on the lower mandible. Further, β-CP reduced orofacial sensitivity in pulp exposed rats. Furthermore, β-CP also significantly decreased AIF and CCL2 gene expression in trigeminal ganglia on the ipsilateral side compared to vehicle treated controls that also received molar drilling. Although these preliminary studies show a beneficial effect of β-CP for the mitigation of orofacial/dental pain, this reviewer strongly feels that the impact of the findings can be significantly enhanced by addressing the following comments.

1. The studies will significantly benefit from the inclusion of additional treatment groups administered a cannabinoid receptor 2 agonist and a PPARg agonist. The lack of these groups limits mechanistic insights.

2. No histopathological findings in the trigeminal ganglia are provided to validate RT-qPCR findings on proinflammatory mediators.

3. It is recommended and customary to use at least two or three endogenous controls and select the best two for RT-qPCR data normalization. Not clear what the endogenous control Rn45s stands for.

4. Since whole trigeminal plexus was used for RT-qPCR studies, the identity of the specific cell types in this structure that contributed to the high AIF and CCL2 expression remains unknown. Immunofluorescence analysis using suitable primary antibodies coupled with cell specific markers followed by quantification of signal intensity is a better approach to quantify inflammatory gene expression. It is possible that expression of other markers that failed to show statistical significance may be differentially expressed in neurons or glial cells or endothelial cells. This could have been captured using immunofluorescence.

Author Response

Thank you so much for the thoughtful review. Regarding additional treatment groups, we plan to initiate these studies next, commencing this summer, including whether CB2 or PPAR antagonists block the effect of BCP, and testing other select agonists. We have preliminary data showing that a selective CB2 receptor agonist O-1966 also works in the assay, but the timepoints tested were different and we did not harvest TG at the time, so those data are not included. We also agree that IHC will increase our understanding of the neuroinflammatory mechanisms at play for BCP but we currently do not have any more tissue left from these animals.  Currently we are months out from having these data. Rn45s is the gene for 45S pre-rRNA transcript. 

Reviewer 2 Report

The study  by Lake et al shows relevant data about the cannabinoid control of nociception. Overall, the manuscript is well presented and the results seems to indicate that cannabinoid drugs could be used to treat orofacial pain. 

My only concern is about Figure 1.

First, I would suggest to present first the Sham experiments and then the Drilled. Second, shouldn't be the value in the baseline condition equal in both graphs (Drilled and Sham)? 

Could the authors clarify this point? This is crucial for the rest of the experiments. 

Author Response

We observed that the baseline sensitivity varied by batches of rats. We actually ran two separate batches of sham rats and their baseline values were lower each time. Because of the time that it takes to perform the testing, the data had to be collected over time using more than one batch of rats. We show the raw data in Figure 1 to be transparent about the batch differences before showing the rest of the data as percent baseline. In future studies we will counterbalance treatment groups following baseline assessments. 

Reviewer 3 Report

I was pleased to review the manuscript “biomolecules-2360378” entitled “Non-psychoactive cannabinoid modulation of nociception and inflammation associated with a rat model of pulpitis” for  Biomolecules. The study aimed to investigate the effects of non-psychoactive Cannabis constituents in the treatment of dental pain and related inflammation. The researchers used two non-psychoactive Cannabis constituents, cannabidiol (CBD) and β-caryophyllene (β-CP) on Sprague-Dawley rats with pulp exposure. The results indicated that β-CP but not CBD produced a significant reduction of orofacial sensitivity. Furthermore, β-CP reduced the expression of the inflammatory markers AIF and CCL2, while CBD only reduced AIF expression. The manuscript shows an adequate design with robust methodologies. Minor revision is advised:

The introduction addresses the topic thoroughly, but the last sentences give many details that are unnecessary in this section. I suggest moving to materials and methods if appropriate or removing them. 

2.1. – How were the animals anesthetized with isoflurane (inhalant) and performed pulpotomy at the same time?

Move sections 2.3. to 2.2, and 2.2 to 2.3.

“It is unclear within our study whether systemic administration of β-CP prevented local inflammation at the site of the tooth pulp” This is quite interesting; if you still have the samples, I strongly advise doing some immunohistopathological evaluation of the periapical area.

I suggest including a paragraph discussing the future perspectives of the investigated substances for clinical use since it's not available worldwide and legally allowed to use.

Author Response

Thank you so much for the thoughtful review. We recline the rats at an approximate 45 degree angle and place a mobile nose cone over the nose of the rat and are at the same time able to open the jaw and proceed with the drilling. Rats are obligate nose breathers so anesthesia is maintained without the cone covering the mouth. We do not have more tissue left but we definitely plan to study the periapical area using IHC, thank you for the suggestion. We are also happy to add commentary regarding the compounds. BCP is not unique to cannabis and can be sourced from other products such as clove, black pepper, and several herbs. It is a recognized as safe food product by the US FDA and is available in nutrition stores. 

Round 2

Reviewer 1 Report

The authors have addressed most of the concerns by stating that they will or are already addressing them in their ongoing or future studies. However, the concern about using a combination of endogenous controls rather than just one as a normalizer for RT-qPCR studies was not addressed. However, this is an important concern that the authors need to be aware of in their future RT-qPCR validation studies.